# Mitochondrial DNA Survey Reveals the Lack of Accuracy in Maremmano Horse Studbook Records

**DOI:** 10.3390/ani10050839

**Published:** 2020-05-12

**Authors:** Andrea Giontella, Irene Cardinali, Hovirag Lancioni, Samira Giovannini, Camillo Pieramati, Maurizio Silvestrelli, Francesca Maria Sarti

**Affiliations:** 1Department of Veterinary Medicine—Sportive Horse Research Center, University of Perugia, via S.Costanzo 4, 06123 Perugia, Italy; camillo.pieramati@unipg.it (C.P.); maurizio.silvestrelli@unipg.it (M.S.); 2Department of Chemistry, Biology and Biotechnology, University of Perugia, via - Elce di Sotto, 8, 06123 Perugia, Italy; irene.cardinali@unipg.it (I.C.); hovirag.lancioni@unipg.it (H.L.); 3Department of Agricultural, Food and Environmental Sciences, University of Perugia, Borgo XX Giugno, 74, 06121 Perugia, Italy; samira.giovannini@gmail.com (S.G.); francesca.sarti@unipg.it (F.M.S.)

**Keywords:** pedigree analysis, Maremmano horse breed, mitochondrial DNA control region, maternal lines, Studbook mistakes

## Abstract

**Simple Summary:**

Breed management and conservation are based on documents (pedigree) that record the relationships among animals although they are not always complete and error-free. Thanks to the DNA analysis, it is possible to check the correct genealogy and breed history. Our study focuses on the Italian Maremmano horse breed, whose Studbook was established in 1980. Starting from a selection of 74 maternal lines, we analyzed the mitochondrial DNA (which is transmitted to offspring only by mother) of 92 samples and verified the pedigree data of 12 dam lines by matching genealogical and mitochondrial information. We found some mistakes in three maternal lines, as the samples belonging to the same lineage showed different mitochondrial DNAs, thus suggesting that the information recorded in the Studbook is wrong and the samples do not descend from the same dam founder. With this research, we confirm the utility of combining genealogical and historical information with molecular techniques in order to prevent errors in pedigree data and the loss of the genetic diversity of local breeds.

**Abstract:**

The Maremmano horse is considered one of the most important Italian warmblood breeds which originated from an ancient population. In 1980, the National Association of Maremmano Breeders established the first Studbook that recorded 440 dams and four sire founders. In this study, we selected the most significant maternal lines in terms of offspring (for a total of 74 lineages and 92 Maremmano horses) and analyzed their mitochondrial DNA control regions. We found a high variability, reflecting the importance of this local breed as a genetic resource to be preserved. Through multiple sampling, we then verified the pedigree information for 12 dam lines by matching genealogical data with mitochondrial haplotypes. A complete concordance was demonstrated in nine lineages, while for the other we highlighted a different number of haplotypes for each line (three in Fiorella, two in Nizza I, and two in Pomposina), thus suggesting that the information recorded in the Studbook could be wrong and the samples do not descend from the same maternal founder. Our combined analysis provides the opportunity to confirm the ancestry of animals and could be employed to prevent errors in pedigree data also for other breeds and species.

## 1. Introduction

The set of known parent-offspring relationships in a population is called pedigree, which is often graphically displayed as a family tree diagram and largely employed to derive the relationships among individuals. Over the decades, pedigree has been used in breeding programs as a predictive tool for the unique and reliable identification of individual organisms in breeding management and conservation [1,2,3]. Pedigree data were already employed to analyze the population structure in order to identify factors that affected the genetic variability of horse [4,5] and cattle [6] breeds. These analyses could prevent the loss of genetic diversity, but in the case of small populations, they could be hindered by limited economic resources, so the accuracy of Studbook data represents a crucial factor for the preservation and management of native breeds.

Each animal should get a unique identification number at birth, and its parents should be known without any doubt. Extensive pedigree records should trace the whole breed history since its establishment; unfortunately, no pedigree is complete and error-free as records are scarce because of different factors:

(1) cut-off dates for recording the genealogies (e.g., from 1950);

(2) animals of unknown origins are included in the Studbooks (e.g., from another breed or country);

(3) exchanges of young animals, parents, or semen at mating, and uncontrolled mating;

(4) wrong registration and administrative mistakes;

(5) fraud.

In this scenario, the development of molecular techniques has a key role in supporting the traditional pedigree which is based on relationships among animals. A combination of both pedigree and molecular data has been demonstrated to be optimal [7,8,9,10]. Due to the lack of recombination and its uniparental inheritance, mitochondrial DNA (mtDNA) is extensively used to genetically distinguish one dam line from another and to trace ancestry through the maternal lineage of a pedigree. Głażewska et al. [11] have demonstrated the usefulness of pedigree data in genetic diversity studies, allowing a focused animal sampling and rare haplotypes identification. On the other hand, many scholars verified the reliability of the maternal lineages recorded in the Studbooks through mitochondrial diversity surveys [10,12,13,14,15,16,17]. The combination of both sources has allowed an extensive phylogenetic reconstruction of domestic horse origins [18,19,20,21,22].

The present study focused on the Maremmano horse, which is considered one of the most important Italian horse breeds for leisure and sports. This warmblood horse descends from an ancient local population spread in Maremma, an Italian territory along the Tyrrhenian coast of Southwestern Tuscany and Northern Latium, located about 100–200 km northwest of Rome. During the Etruscan period and over the centuries, the Maremmano horse was influenced by various genetic types introduced by people that dominated this territory (more details in [23,24,25]). Even though after the first half of the 19th century the Maremmano was strongly crossed with the Thoroughbred and other worldwide breeds, it has preserved its noteworthy genetic diversity [26,27]. After the Second World War, the number of Maremmano horses dropped (from almost 12,000 surveyed units in 1940, decreased to about 5000 surveyed units in 1950 [23]), although in the 1970s a new interest in this breed rose due to its versatility in adapting to different roles: stock, farm, and sport. In 1979, a conservation program started thanks to the creation of the National Association of Maremmano Breeders (ANAM) and the establishment of the Studbook in 1980. Nowadays, it registers records of 2251 mares and 107 stallions (mostly bred in the provinces of Grosseto and Viterbo, in Central Italy).

Although a significant proportion of Maremmano horses descends from a strongly limited number of male lines (each living horse is officially assigned to one out of four lines [2]), it is evident that multiple dam lineages contributed to the extant genetic variability, especially in terms of mitochondrial diversity [21,27]. The origin of Maremmano’s founding mares is very heterogeneous and has to be searched not only in native dams that populated Central Italy between the late VIII and early IX centuries (for which the only unchanged condition during time was the wild environment in which the horse lived) but also in occasional introgressions from other breeds, as testified by historical documents for Another Delight and Blanca lines (Thoroughbred and Lipizzan partbred, respectively). Furthermore, out of the 440 founding mares (born between 1920 and 1987), the majority of them (N = 332 mares) were born between 1960 and 1975 [2].

In this complex scenario, our research aims to match the pedigree data available for the Maremmano breed with mitochondrial haplotypes in order to assess its genetic diversity and verify the reliability of current genealogies, as phenotypic traits and genealogical data are insufficient to ascertain the horse history and the real pedigree.

## 2. Materials and Methods 

### 2.1. Samples

Genealogical information recorded in the Maremmano Studbook was considered in order to choose all principal dam lines and select samples with a known pedigree up to four consecutive generations of maternal ancestors. Generation intervals (GI) were calculated as the difference between the average age of offspring and parents. Also pedigree completeness (PEC) was calculated including maternal and paternal lines. The genetic distance between horses was evaluated by calculating the average relatedness (AR) coefficient (twice the mean coancestry between a given animal and all animals in the population including itself, which also considers the inbreeding coefficient) and only animals with at least 20 descendants were selected for this study. Pedigree data were obtained from the National Horse Breeders Association of Maremmano Breed (ANAM). These data are based on old documents (related to sales and mating registrations) which are often incomplete and recorded up to the subject arbitrarily considered the breed-founder. For this reason, the data were verified by referring to the archives of the Sports Horse Research Center (University of Perugia), that were created thanks to the access to the main national Studbook databases for scientific purposes and allowed the cross-checking of genealogical information which are integrated, where it is possible, with molecular genetic techniques.

A total of 92 Maremmano horses (60 females and 32 males) belonging to 74 female lines from different Italian regions (Emilian and Ligurian Appennines, Tuscany, Umbria, and Latium) were analyzed (Appendix A).

Among them, due to the sampling complexity as these horses are bred in the wild, we were able to verify the pedigree information through the mtDNA analysis of 12 of the most important maternal lineages in terms of offspring, by analyzing two or more samples for each dam line, for a total of 30 samples (18 females and 12 males). They belonged to the four main male lines Aiace (N = 12), Ingres (N = 2), Otello (N = 13), and Ussero (N = 3) and were representative of all the four Italian regions mentioned above.

### 2.2. Pedigree Analysis 

For each animal, the pedigree reported in the ANAM Studbook was checked for possible mistakes and arranged for the analysis by using the Pedigree Viewer v.6.3 software [28], which is able to read and analyze the entire pedigree data, check for logical errors, and organize the animals in a chronological order of generations. The generation intervals (GI), pedigree completeness (PEC), inbreeding coefficient (F), and the average relatedness (AR) coefficient were computed using “Endog v 4.8” software [29]. For each horse in the whole Maremmano population, the percentage of Thoroughbred blood and the reconstruction of dam and sire ancestors’ lines were carried out using an in-house Fortran software and graphed using Microsoft Excel.

By using a script in Fortran 95 software, it was possible to calculate the percentage of Thoroughbred blood for each animal. The algorithm computes the Thoroughbred blood percentage reported in the pedigree through logical considerations: the blood percentage of the animals identified as Thoroughbred is 100%. Each horse, whether male or female, transfers to the offspring half of its own coefficient so that the product resulting from mating two Thoroughbred has 100% Thoroughbred blood percentage, which is inherited half from the sire and half from the dam. Therefore, an individual horse who has just one Thoroughbred parent has 50% Thoroughbred blood whereas an individual horse with one Thoroughbred grandparent has 25% Thoroughbred blood.

The software computed the Thoroughbred blood percentage in Maremmano horses as
(1)$$\;{\left(\frac{1}{2}\right)}^{n{}^{\circ}generations}=\mathrm{Thoroughbred}\;\mathrm{blood}\;\mathrm{coefficient}$$

The average per year of birth of these Thoroughbred blood coefficients was reported in a plot in order to analyze the trend of the Maremmano breeding program. 

### 2.3. Reconstruction of Sire’s and Dam’s Lines

Sire and dam ancestors’ lineages of the pedigree were reconstructed by using Fortran 95 software, which allowed us to trace backwards the maternal and paternal lineages of each individual pedigree till the ancestors without genealogical information which are supposed to be the breed founders. A preliminary selection of the animals was carried out by employing this software in order to perform a molecular analysis of Y chromosome (only for male individuals) and mitochondrial DNA. After specific logical tests, the program returns an output file containing the name of each individual with the relative male and female ancestors back to the founder and indicating the number of generations that separates them. For each male and female founder, the total number of individuals belonging to its lineage, as well as the number of living and still in activity animals was computed.

### 2.4. Mitochondrial DNA Control Region Analyses 

After the pedigree analysis, only animals with at least 20 descendants were selected for this study, for a total of 92 individuals belonging to 74 dam lines. Among them, the pedigree information was verified for 30 samples belonging to 12 of the most important Maremmano dam lines in terms of offspring (Adua III, Arpa 3, Balilla, Carmela, Cecilia, Fiorella, Germana, Giulia, Mora (3484), Nizza I, Pomposina, and Velia), by analyzing 27 mtDNA control-region sequences from Cardinali et al. [27] and three additional sequences here published for the first time (Appendix A).

For these three sequences, DNA was extracted from 1.2 mL of peripheral blood samples by automated extraction through the MagCore^®^ Automated Nucleic Acid Extractor, following the provided protocol. All experimental procedures were reviewed and approved by the Animal Research Ethics Committee of the University of Perugia (Prot. N.2017_01).

Mitochondrial DNA control-region was amplified between sites 15,364 and 563 and Sanger-sequenced as already reported by Cardinali and colleagues [27]. Fragments of 610 base pairs (from nucleotide positions 15,491 to 16,100) resulting from the standardization of the sequencing were aligned using the software Sequencher™ 5.2 (Gene Codes Corporation, Ann Arbor, MI, USA). The mutational differences in relation to the equine reference sequence (ERS: NC001640, derived from X79547) available in GenBank were registered to allow the haplotype identification; the mutational motifs were used to classify all the mtDNAs into their respective haplogroups [21]. The phylogenetic relationships among the 92 Maremmano samples were visualized through the construction of a median-joining tree using Network 10.0 software (www.fluxusengineering.com).

## 3. Results

### 3.1. Pedigree Analysis

The pedigree file, including 16,588 horses (9718 females and 6870 males) born from 1920 to 2019 (30 September 2019) provided by ANAM Studbook, was verified. In the first year of ANAM official activity 203 foals were registered and this number did not change in the following two years. Then, the number increased to 540 in 1991, reaching its maximum in 1997 (557 foals). Subsequently, the number of foals decreased to 370 in 2000 and remained almost stable until 2012 when it decreased to 216 because of the economic crisis. The number registered in 2019 was practically the same as the number registered when the Studbook first started (Figure 1).

The average ± SD Thoroughbred blood percentage in the whole population was 19.33% ± 2.65: 19.81% ± 2.06 in males and 18.92% ± 2.71 in females (Figure 2). 

It has to be pointed out that the percentage of Thoroughbred blood in pedigree has increased 0.19% per year since 1980; however, the increase more than doubled to 0.51% in the first 10 years of the ANAM activity and remained stable in the following years.

The average inbreeding (F) in the whole population was 2.90% and ranged from a minimum of 1.29% (1981) to a maximum of 6.16% (2014); this value increases on average 0.10% per year although it decreased to 4.64% in 2017, due to the execution of a Breeders Association inbreeding containment program started in 2012 (Figure 3).

The average inbreeding increases to 4.54% in the inbred animals while this value reaches 3.92% in the collected sample.

The mean of the AR coefficient in the whole population was 5.17 % and ranged from a minimum of 0.01% to a maximum of 14.62%.

### 3.2. Reconstruction of Sire’s and Dam’s Lines

The average generation interval was 10.65 years; the greater pathway was observed between the Maremmano sire and foal (11.64 years), while the lower (9.65 years) was between the mare and foal. The average PEC was 99.34% including maternal and paternal lines.

The sire lineage, identified by the “home made” software for the whole Maremmano population, confirmed four main lines (Aiace, Otello, Ingres, and Ussero), and smaller lines with a restricted living progeny. As for the dam lineages, a selection based on not related lines with at least 20 descendants and living progeny was carried out. The use of this procedure permitted to identify 74 maternal lineages, among which there are the most famous dam founders: Arpa 3, a mare born in 1974, whose progeny is part of the unit of Tenuta Presidenziale di Castelporziano mares [4]; Carmela, born in 1960; Ilona, a mare of “della Nave” farm; Blanca and Another Delight, already mentioned in the section Materials and Methods as the founders of the Maremmano breed.

### 3.3. Mitochondrial DNA Control Region Analyses 

From the individuals belonging to the 74 maternal lineages, 92 animals were both tested and sampled as described in the Material and Methods section, on which a mitochondrial DNA analysis was carried out. The 92 individuals showed an average AR coefficient of 3.57 with a minimum of 0.01% and a maximum of 12.23%. Among the 610 bp analyzed, 75 polymorphic sites were found, all of them represented by single nucleotide polymorphism (SNP) with an average nucleotide diversity of (π) 0.02063 [25].

Fifty-one different haplotypes were identified (Nh), with an average haplotype diversity index (Hd) of 0.981 [30], showing a high mitochondrial diversity among the Maremmano samples. The analysis of the 75 mutated sites defined 35 haplotypes specific of one dam line and 16 shared by different maternal lineages which likely testified for a common phylogenetic ancestor; only one sample (MA334) did not show any differences compared to the reference sequence (ERS) (Appendix A). 

A total of 16 (A, B, C, D, E, G, H, I, J, L, M, N, O, P, Q, and R) haplogroups were identified, with the higher frequencies for L, G, B, and M (20.65%, 17.39%, 14.13%, and 13.04%, respectively) (Figure 4 and Appendix A).

After the comparison between the genealogical data from the pedigree and the molecular data from the mtDNA analysis, a total of 12 dam lines were checked, as it was possible to confirm if two or more samples belonged to the same maternal line (Appendix A). In our study, Fiorella, Nizza I, and Pomposina lines showed different haplotypes within the same line. This finding may suggest that the information collected in the studbook could be wrong, and the samples do not descend from the same maternal founder. In particular, Fiorella showed three different haplotypes which were not shared with other lineages; samples from Nizza I had two haplotypes, one of which was shared with Patrizia and Pomposina lineages, while the other haplotype from Pomposina was shared with Serenella lineage.

## 4. Discussion 

Knowledge of relationships among horses is essential for the genetic management of breeds and represents one of the principal tools employed to optimize the conservation strategies. It is traditionally retrieved from pedigree, which is essentially never complete or error-free. With the development of molecular techniques, traditional relationship coefficients based on pedigree information are gradually replaced by those calculated from molecular data. In this framework, the present study combined the available pedigree data from the Maremmano breed with mitochondrial DNA haplotypes, in order to assess its genetic diversity and verify the reliability of some genealogies. This uniparental molecular marker can be applied to historical issues of maternity in absence of biological material from putative dams or even from offspring [12]. The mtDNA diversity previously described for this breed [21,27] testifies for a multiple dam lineage contribution, whose legacy should be searched also in the occasional introgressions from other breeds, as reported by historical documents for Another Delight and Blanca lines.

Our study confirms the completeness of genealogical information and the traditional importance that breeders gave to the female lines; these two factors had a key role in preserving the genetic diversity of the Maremmano horse. Thanks to accurate genealogical analysis, 16,588 Maremmano horses registered in the studbook were better identified, and male and female ancestors were traced. From the reconstruction of sire and mare lineages, it was confirmed that all Maremmano horses derive from 4 males and 440 dam founders, of which 332 mares were born between 1960 and 1975 [2]. The value of average generation interval (10.65 years) highlighted the slow development of this breed. The greater distance (11.64 years) occurred on the pathway stallion-foal indicated a more severe selection of young stallions according to the ANAM breeding plans. The opposite situation was showed by the pathway mare-foal, where the distance was lower (9.65 years). This finding could indicate that a high number of young mares was used for breeding, while there was not a quick withdrawal of breeding stallions.

Among them, the most significant maternal lines were selected for a total of 74 lineages. When combined with genealogical and historical information, mtDNA data proved to be helpful in evaluating maternal inheritance and in identifying founding mares. Furthermore, mtDNA surveys contributed to the current understanding of the domestication process [18,19,20,21,22] and tracked breed migration and distribution by comparing maternal lines among different populations.

For these reasons, we collected blood samples from 92 Maremmano horses belonging to the 74 dam lines which were considered the most representative of this local breed (at least from the genealogical records). Among them, a molecular concordance with studbook records was demonstrated through multiple sampling in nine lines (Adua III, Arpa 3, Balilla, Carmela, Cecilia, Germana, Giulia, Mora (3484), and Velia), while genealogical mistakes emerged for three dam lines (Fiorella, Nizza I, and Pomposina), probably due to wrong data registration. Our findings confirm the importance of the pedigree reliability for the analysis of genetic structure, that could be applied to other local breeds and species, as previously done [4,5,6].

The analysis of Maremmano mtDNA control regions revealed a high mitochondrial variability within the breed, testified by 75 mutated sites that allowed to identify 51 haplotypes and 16 different haplogroups, thus confirming its remarkable genetic diversity [26,27]. Among the 51 haplotypes, 16 were shared between different lines, while the other 35 were each present in one lineage. This indicates that even if the genealogical data should be reliable and probably without mistakes, they could lead to wrong conclusions because the pedigree data might be temporarily restricted, and different maternal lineages could share the same haplotypes. It should be noted that the mtDNA mutation rate for horses is about one mutation every 1358 years [21], thus the same haplotype shared by different lineages could have been differentiated more than 1000 years ago, identifying a common founder for different maternal lines.

## 5. Conclusions

The development of molecular techniques gradually sustains traditional pedigree-based analysis; a studbook is not always error-free therefore mtDNA analysis can provide genetic information about the maternal lines. Our study showed that the combination of both sources not only could be optimal but could also enable a phylogenetic reconstruction of domestic horse origins. The mtDNA analysis provides breeders with the opportunity to confirm the ancestry of their horses and could be used to prevent possible frauds in falsifying the genealogical data. 

Furthermore, through the analysis of the entire mitogenome and the comparison with other breeds, it will be possible to characterize all the different haplogroups existing in the Maremmano breed and incorporate this scientific approach in the selection plans, in order to provide important support to preserve its genetic diversity.

In conclusion, the present research confirms the power of mitochondrial DNA surveys for testing pedigree authenticity and improving the monitoring and management of livestock populations and demonstrates the utility of mtDNA analysis in describing the maternal legacy among samples by excluding those maternally related or without certain origins. This research can be considered as a preliminary study for further analyses focused on verifying the reliability of pedigree data for the remaining Maremmano dam lines and can be also extended to other local breeds and species.

## Figures and Tables

**Figure 1 animals-10-00839-f001:**
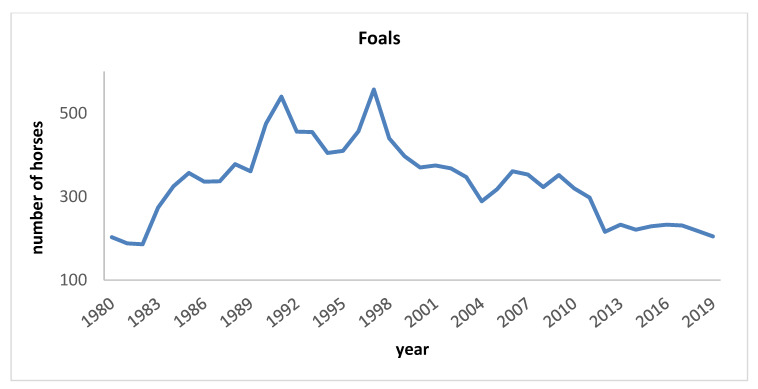
Number of Maremmano foals per year from 1980 to 2019.

**Figure 2 animals-10-00839-f002:**
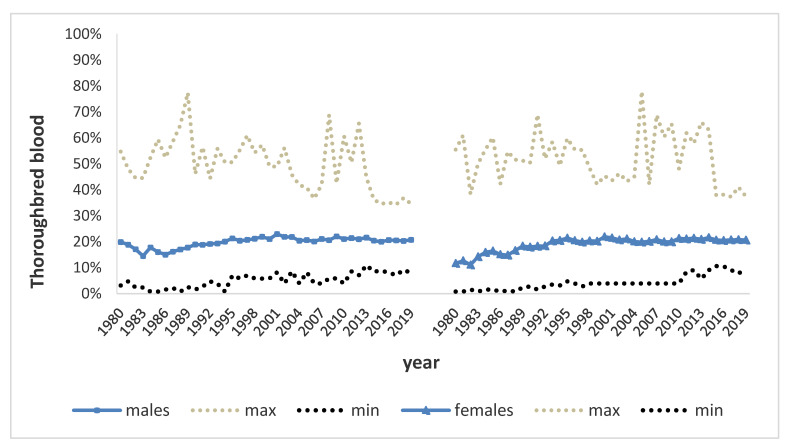
Maximum, minimum, average Thoroughbred blood percentage in the whole Maremmano population per year of birth from 1980 to 2019.

**Figure 3 animals-10-00839-f003:**
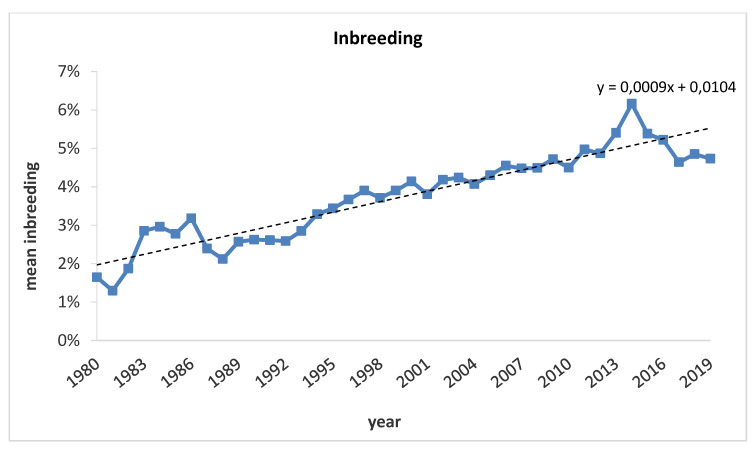
Average inbreeding per year for each Maremmano horse here analyzed.

**Figure 4 animals-10-00839-f004:**
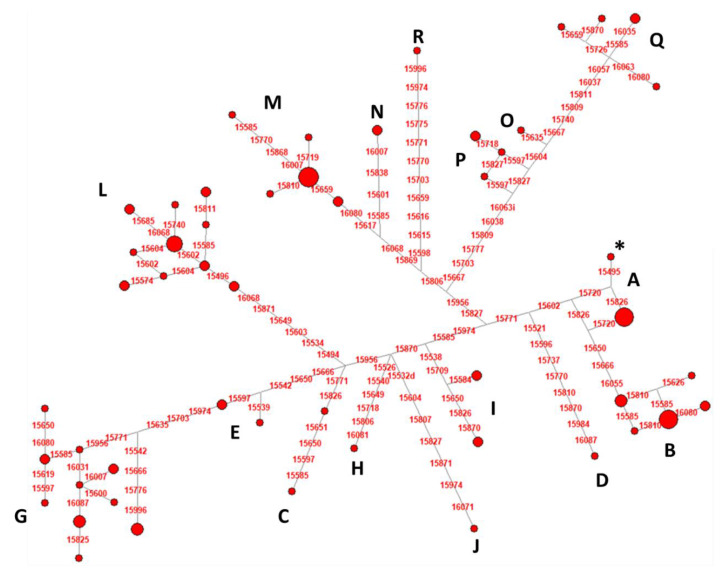
Median-joining network based on the control-region sequences (nps 15,491–16,100) of 92 Maremmano samples. The asterisk indicates the haplotype identical to ERS (Equine Reference Sequence; NC_001640.1). The figure shows the genetic relationship between haplotypes where each SNP indicates a single mutational event; each red circle is proportional to the number of samples carrying the same haplotype, and haplogroups are indicated with capital letters (see Appendix A for more details).

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
