# Peer review of "Mitochondrial DNA Survey Reveals the Lack of Accuracy in Maremmano Horse Studbook Records"

_animals, 2020, doi:10.3390/ani10050839_

Round 1

Reviewer 1 Report

The paper deals with the analysis of the pedigree of the Maremmano horse breed and the possibility of reducing any errors through the analysis of the mtDNA. The work looks good, although the discussion must be completely revised, as it currently reports only results and does not analyze the bibliography. A figure does not report the correct data and part of the M&M should be better explained.
Furthermore, some small corrections are shown below:

L15 and L20: the term “Studbook” should be preferred instead of “Genealogical book”

L22 and L35: not only frauds are the causes of failure to combine genetic and pedigree data, as stated at lines 49-54 (points 1-4).

L67-69: this is a conclusion and should not be included in Introduction

L106-108: why data of Studbook were verified? If the Studbook is the official registration of genealogies, any other registration should be already verified and not allowed. Please explain why the Sport Horse Research Centre data should be more correct than the Official Studbook.

L124, L127: the use of home-made software is possible but a validation with proved official software should be reported and referenced. Some of the cited parameters are obtained also from software largely spread in the scientific context (ENDOG for example, with reference to the measure of the inbreeding level for each animal, L128, and parameters from L129 to L132) and the use of homemade software is not justified.

L156: it’s not clear the way you choose the 12 dam lines; why 12? At L109 you say they are 74, so it is not clear how they reduced to 12. At L220 authors say that “Out of the 74 lines, 12 matched with the pedigree information”, does it means that the remaining 62 do not match? The authors should explain better how the 12 dam lines were selected.

L164: use commas to separate thousands, as correctly done at L166. Furthermore, I am not sure that what is reported in L164 (“sites 15364 and 563”) is correct: it is probably more correct to write "sites 15,364 and 15,563).

L174: See what has been written before in relation to the need to verify the data. In any case, this sentence is redundant and is not a result but a method.

L175-180: the data reported in the text are not the same reported in the figure (for example 557 foals in 1997 is not in the figure). Please check the adherence of the data in the text to the figure.

L182: the data reported are not “average” but “average ± SD”.

Figure 2: the graph could be improved by adding the limits (minimum and maximum) of the percentage of thoroughbred blood for each year. Probably (but this is not mandatory), an analysis of variance of the effect of year on percentage of thoroughbred blood in Maremmano horse could be useful to assess the develop of the breed;

L189-191: the drop reported from 6.16% to 4.64% from 2014 to 2017 should be justified: implementation of a support to breeders to avoid mates that generate too inbred foals?

L204: DeLight or Delight as at L89?

L208, 209, 211, 213: do not use commas for decimals.

L240-262: these are results and are already been reported in the appropriate section. The discussion is too poor; it must be greatly improved through the analysis of the bibliography. In this regard it can be advised that a part of the introduction (for example from L83 to L92) could be included in the discussion.

Reviewer 2 Report

Mitochondrial DNA survey reveals the lack of accuracy in Maremmano Horse studbook records by Giontella et al. 

This study tracked the genealogy of the Maremmano horse breed using studbook records (including morphological traits) and mt DNA. The study is interesting and has merit but there are a few issues that need to be resolved before publication. 

The title needs to be changed. First of all, I think it needs to change "accuracy" by "accuracies". However, this does not mean anything at all. Lack of accuracies on what? In dam founders.... I think this part is missing.

Line 14: what do you mean with "real" genealogy?

Line 42: please change "among animals" by "among individuals"

Line 43: please consider to change "animals" by "individual organisms with their ancestry"

Line 49 and 50: delete "(e.g. 1950)" and (e.g., from......)"

Lines 65-69 can be moved to the last paragraph of the Introduction

Lines 72-73: the parenthesis can be removed and start a new sentence with "During the Etruscan......"

Line 78 and 83: what do you mean with "shortly" and "strongly".

Line 79: delete "horse" in "sport horse"

Line 87: I'm not familiar with the date description '800 or '900. Can this be changed to centuries or years as AD? 

Line 96: At the end of the Introduction there is a statement about the "morphometric approach" but nothing about relevant morphological traits in this breed is found in the Intro. More information about morphological characteristics is needed there.

Line 102: How many years are four generations? Did you take four consecutive generations? or did you allow missing data among the four generations? More explanation is needed here.

Line 103: explain "AR"

Line 111 and somewhere else: is this the right way to cite the tables in the ms according to the journal guidelines? Tab. 1?

Table 1: This table needs to be organised in a different way. Right now is organised by "Sample ID" but this doesn't mean much to the reader. So I suggest the table can be organised by "Sex" or "Maternal line". 

Line 114: what do you mean by "12 of the most important....."

Lines 182-183: add "%"

Line 213: high diversity on the tested samples?

Line 223: this is a redundant argument I suggest deleting it. 

I would like to see a phylogenetic tree or network from the mt data showing haplogroups as in reference 18. 

Most of the paragraphs in the discussion section can go to the Intro, Analysis or Results sections. Also, I would like to know more about what are the most likely source of the errors and where the errors probably come from?

Reviewer 3 Report

I revised the paper “Mitochondrial DNA survey reveals the lack of accuracy in Maremmano Horse studbook records” by Giontella et al. The paper is very interesting and have potential to be relevant for Animals readers. However in the present form the introduction and discussion are a bit weak. As an example, they do not provide much information on the importance of correctness of pedigree data and especially in the discussion a paragraph about future exploitation of the results in missing. Few more editions are also needed in the material and methods and results section to allow a better flow throughout the manuscript.

Please find below my detailed comments:

Abstract

Line 26 recorded

Line 27 Can you specify what you mean with “most significant lines?” are the most representative ones in terms of offspring? Please specify better.

Line 28 “testifying” replace with reflecting the importance of this…

Line 31 Can you specify if the concordance for the nine lineages was complete or partial?

Line 31 instead of “some mistakes” can you replace with the exact number e.g. the percentage of uncorrected assignment?

Introduction

General comment 

In the present form, the introduction does not emphasize the importance of pedigree data and thus the importance of their correctness, especially in small native population. Analyses at pedigree level are still effective tools to unravel the state of genetic diversity, especially when studying small and under-development populations. This kind of populations have limited economic resources which cause often the unavailability of more advanced technologies (e.g. genomics). Exploring genetic diversity at pedigree level does not need extra economic resources, since from already-available data it is possible to identify indicators of genetic variability. For this reason, I think it would be great to stress a little bit more the importance of the correctness of those type of data. You can find several recent examples in the literature where pedigree data are used to assess genetic variability. As an example

In horses:

  • Ablondi et al (2018) https://doi.org/10.1111/jbg.12357
  • Vostrá-Vydrová et al (2016) https://doi.org/10.1016/j.livsci.2016.01.001

Other species:

  • Fabbri et al (2019) https://doi.org/10.3390/ani9110880

Minor comments:

Line 67: replace extant with more appropriate term

Line 67 to Line 69 to me seems more discussion/conclusion of your study. Please place those sentences in a better place throughout the manuscript.

Line 78 please replace specimens with horses or animals

Line 88 lives to lived

Material and Methods

Table 1  can be a supplementary material and can be replaced with a bit more data description in the text. For example: how many horses belonged to same paternal lines and region of belonging. 

L118 You mentioned that you had information about coat color and biometric measurements but then you do not present any results about those characteristics.

L122 “for each horse in the population” do you mean the 92 or the whole population? Please specify better the number of horses used for this analysis

L121 replace tiers with better word. Perhaps generations?

L134 to L140 is this explanation needed?

L156 can you specify which criteria did you use to set the “most important Maremmano dam lines”? they were the most important for which reason?

L194 please be consistent with decimals separator, change 4,54% to  4.54%  

Results

In general, it would be great to add a bit more information about the pedigree data you are presenting. The average inbreeding in the different generation based on pedigree data is dependent on the pedigree depth. Therefore, it would be good to know the pedigree depth of animals belonging to different generation as I suspect that it will change considerably across time.

Please specify if from line 198 you are still presenting results for the whole database or for the selected 92 horses.

The paragraph 2.3. Reconstruction of sire’s and dam’s lines mentioned in the m&m is not reported in the results neither the paragraph called “ 2.4. Mitochondrial DNA control region analyses”. For a better flow please keep the same paragraphs also in the result section and check that all you mention in the m&m is also presented in the results. E.g.: at L151 to L154 you stated: After specific logical tests, the program returns an output file containing the name of each individual with the relative male and female ancestors back to the founder and indicating the number of generations that separates them. For each male and female founder, the total number of individuals belonging to its lineage as well as the number of living and still in activity animals was computed.”

However I do not see those results in the present form of the manuscript.

L200 If there were 440 maternal lines in total, I would put this number in the material and method and not the already corrected one at L109.Otherwise clarify better those results as are not very clear

L206 did you test 92 animals or 30? At L156 you wrote 30. Please clarify

L220 this sentence is not clear. Just 12 out of 74 lines were confirmed by mitochondrial dna? But did you not test 30/92 animals descending from 12 lines? Please clarify, I would suggest replacing “matched” with a better verb as it is misleading. If I understood correctly what you meant, I would suggest replacing “matched” with: “the 12 maternal lineages tested in this study were all confirmed yet for three of them with some degree of discrepancy ….

Discussion

In general the discussion is well written. However, it can be improved adding some more points of discussion. As stated above in the “introduction”, more discussion about the importance of pedigree data and their correctness would be appreciated. Also, some more discussion about possible future studies could be nice. You have stated them partially in the conclusion, but it would be better to have a paragraph as well in the discussion.

Round 2

Reviewer 2 Report

Hi,

Please find my comments within the pdf attached. Please take special care of my suggestion about extended explanations in Figure 4. 

With best wishes (especially good health in these difficult times).

Regards,

Author Response

  • L14: we deleted “the”.
  • L65-70: we moved “Pedigree data were already employed to analyse the population structure in order to identify factors that affected the genetic variability of horse [4, 5] and cattle [6] breeds. These analyses could prevent the loss of genetic diversity, but in the case of small populations, they could be hindered by limited economic resources, so the accuracy of Studbook data represents a crucial factor for the preservation and management of native breeds” at the end of the first paragraph of the introduction.
  • L71: we delete “In this framework,”
  • L131: we modified the sentence as follows: “….order of generations. The generation intervals….”.
  • L133: we started a new sentence after “Endog v 4.8” software [29]” and deleted “while”.
  • L171: we deleted “(from np 15,491 to np 16,100)” from the Table S2 as the correct information is already reported in the title of the Table.
  • Figure 1,2 and 3: to be improved as suggested by the reviewer
  • L224: the 75 polymorphic sites are all represented by SNPs (haplotype) reported in Table S2. Thank you for the suggestion, but we think that it could be very difficult to use different colours for each maternal line (N= 74) in Figure 4, as the samples are 92 in total. Table S2 reported in detail all maternal lines, haplotypes and haplogroups and it could be considered as a description of the Network tree. However, we improved the caption of Figure 4 as follows: “Median-Joining Network based on the control-region sequences (nps 15,491-16,100) of 92 Maremmano samples. The asterisk indicates the haplotype identical to ERS (Equine Reference Sequence; NC_001640.1). The figure shows the genetic relationship between haplotypes where each SNP indicates a single mutational event; each red circle is proportional to the number of samples carrying the same haplotype, and haplogroups are indicated with capital letters (see Table S2 for more details).”.
  • L228: we modified the sentence as follows: “The analysis of the 75 mutated sites defined 35 haplotypes specific of one dam line and 16 shared by different maternal lineages which likely testified for a common phylogenetic ancestor; only one sample (MA334) did not show any differences compared to the reference sequence (ERS) (Tab. S1)”.
  • L257: we modified “contribute” with “contribution”.

With best wishes tread carefully

Reviewer 3 Report

The manuscript has improved significantly after the first round of revision, I really appreciated the work done by the authors. I still have few minor comments that you can find below:

Table 1: Sorry but I am still not fully convinced that this table should be published in the present form. It is a 3 pages long table and I think it would be better to present those data either in the supplementary materials or in another form. A possible suggestion is to make plots based on the data presented (e.g. pie plot for the sex, histograms for the paternal line counts and region of belonging.

L126 could you add a sentence explaining which are the 30 horses from the 92 horses that were then analyzed? E.g. the 30 horses further analyzed were xxx female xxx male belonging to xx paternal lines and located in xxx regions

L203 replace 4,54 to 4.54%

L220 I think you should replace here 92 with 30 horses

L235 the figure 4 legend is not placed correctly in the manuscript

Author Response

Table 1: Sorry but I am still not fully convinced that this table should be published in the present form. It is a 3 pages long table and I think it would be better to present those data either in the supplementary materials or in another form. A possible suggestion is to make plots based on the data presented (e.g. pie plot for the sex, histograms for the paternal line counts and region of belonging.

Thank you for the suggestion concerning Table 1. We decided to move it in the supplementary materials as Table S1.

L126 could you add a sentence explaining which are the 30 horses from the 92 horses that were then analyzed? E.g. the 30 horses further analyzed were xxx female xxx male belonging to xx paternal lines and located in xxx regions

Thanks for your suggestion. We described these 30 samples by modifying this sentence as follows: “Among them, due to the sampling complexity as these horses are bred in the wild, we were able to verify the pedigree information through the mtDNA analysis of 12 of the most important maternal lineages in terms of offspring, by analyzing two or more samples for each dam line, for a total of 30 samples (18 females and 12 males). They belonged to the four main male lines Aiace (N=12), Ingres (N=2), Otello (N=13) and Ussero (N=3) and were representative of all the four Italian regions above mentioned.”.

L203 replace 4,54 to 4.54%

We replaced “4,54” with “4.54%”.

L220 I think you should replace here 92 with 30 horses

Here we referred to all the 92 samples, as we analysed all their mtDNA control-region sequences.

L235 the figure 4 legend is not placed correctly in the manuscript

We moved the Figure 4 legend in the correct position.

Thanks for your support!

Best regards